# Infants’ Sleep: Israeli Parents’ Knowledge, Attitudes and Practices

**DOI:** 10.3390/children8090803

**Published:** 2021-09-14

**Authors:** Anat Shatz, Leon Joseph, Liat Korn

**Affiliations:** 1Shaare Zedek Medical Center, Jerusalem 9103102, Israel; atidbaby1@gmail.com (A.S.); leonj@doctors.org.uk (L.J.); 2Atid-Israeli Foundation for the Study and Prevention of Sudden Infant Death, Jerusalem 9103102, Israel; 3Department of Health Man agement, School of Health Sciences, Ariel University, Ariel 40700, Israel

**Keywords:** safe sleep, parents’ knowledge, attitudes, infants’ sleep practices, sudden infant death syndrome (SIDS)

## Abstract

The purpose of the study was to assess Israeli parents’ knowledge of and attitudes towards practices promoting infants’ safe sleep and their compliance with such practices. Researchers visited the homes of 335 parents in 59 different residential locations in Israel and collected their responses to structured questionnaires. SPSS 25 statistical package for data analysis was used. Attitude scales were created after the reliability tests and scaled means of parental attitudes were compared between independent groups differentiated by gender, ethnicity, and parental experience. A logistic regression was run to predict the outcome variable of babies’ sleep positions. The total knowledge score was significantly higher for women (56.3%) than for men (28.6%; *p* < 0.001). Arabs were more committed to following recommendations (29.3%) than Jews (26.9%; *p* < 0.001). Consistent with safe sleep recommendations, 92% of the sampled parents reported avoiding bedsharing and 89% reported using a firm mattress and fitted sheets. The risk of not placing a baby to sleep in a supine position was higher among older parents (adjusted odds ratio—AOR = 0.36, 95%CI 0.16–0.82), smoking fathers (AOR = 2.66, 95%CI 1.12–6.33), parents who did not trust recommendations (AOR = 4.03, 95%CI 1.84–8.84), parents not committed to following recommendations (AOR = 2.83, 95%CI 1.21–6.60), and parents whose baby slept in their room (AOR = 0.38, 95%CI 0.17–0.88). Knowledge of safe sleep recommendations was not associated with actual parental practices. Trust of and commitment to recommendations were positively correlated with safe sleep position practices. It is essential to develop ethnic-/gender-focused intervention programs.

## 1. Introduction

Sudden infant death syndrome (SIDS) remains the leading cause of infant death between the ages of one month and one year in the Western world [1]. In Israel, the Central Bureau of Statistics publishes data on SUID (sudden unexpected infant death). Rates in the last decade ranged from 0.6–0.23 per 1000 live births. The most recent available reports for the years 2016 and 2017 indicated 40 and 53 cases of SUID, respectively [2]. Despite extensive educational campaigns, parental compliance with risk reduction recommendations remains low [3]. Previous surveys by ATID, the Israeli foundation for the study and prevention of sudden infant death, and the Ministry of Health [4] found that only 33% to 40% of Israeli parents placed their infants to sleep on their back [5], even though this is known to be the number-one risk factor for sudden infant death [1]. Low compliance was also found with respect to other risk factors.

Recommendations regarding what constitutes a safe sleeping environment for a baby are published by Israel’s Ministry of Health and ATID based on the recommendations of the American Academy of Pediatrics [6], and are promoted by healthcare professionals, including doctors and nurses in hospital maternity wards and community health centers (Tipat Halav). These evidence-based recommendations include supine sleeping position, sleeping in the parents’ bedroom, avoiding bed-sharing, avoiding overheating, avoiding exposure to passive smoking, and breastfeeding for at least two months. Other recommendations include avoidance of head covers, a soft mattress, heavy bedding with loose blankets, and crib bumper pads [6]. 

In order to reduce SIDS incidents, it is important to raise awareness and knowledge of safe sleeping environments for infants, increase compliance with the recommendations, communicate what is expected of parents, and provide a positive role model for parents and caregivers [7]. Providing information to raise awareness is a well-known health promotion technique that improves compliance with safety practices and decreases mortality [8]. Risk reduction campaigns such as the “Back to Sleep” campaign that emphasize the importance of avoiding multiple and simultaneous risk factors for SIDS are essential [9]. Knowledge, attitudes, and practices (KAP) regarding various health promotion issues are keys [10] to the goal of behavior change [8] and are the rationale behind many campaigns to raise awareness. The “Back to Sleep” campaign was followed by a dramatic (65%) decrease in the incidence of infants placed prone to sleep [9], emphasizing the strong relationship between knowledge and practice.

In Israel, a prior study in the northern district of Haifa found that Israeli-born Jewish mothers were more likely to place their babies prone than Israeli-born Arab mothers [11]. The number of children in a family was found to be associated with the prone position of the baby, with a significant inverse relationship between number of children and compliance with non-prone sleeping at 2 months old [12]. Gender was also found to play an important role in determining sleep position. Male caregivers demonstrated less knowledge of the recommendations regarding infant sleep safety, and they adhered less to all of the safe sleep recommendations [13].

The aim of this study was to assess Israeli parents’ knowledge of and attitudes towards practices promoting infants’ safe sleep and their compliance with such practices. The current study focused on the question of whether parents’ attitudes towards what constitutes a safe environment could predict adherence to safe sleep recommendations. To this end, we examined Israeli parents’ knowledge of risk reduction strategies, their perceptions of what constitutes a safe sleeping environment for a baby, whether they believe that these strategies are effective, whether they believe that the recommendations are important, and their reports of actual practice. Research questions focused on the differences in knowledge, attitudes, and practices by gender, ethnicity, and parental experience and identifying which of these variables were associated with babies’ sleep position.

Except for the Haifa survey and ATID’s survey from 2006, we could not find any new data on parental behavior regarding safe sleep practices for babies in Israel, hence the importance of this study. The study’s findings will help health care providers identify behavioral barriers and issues that prevent the adoption of safe sleep behaviors by parents and develop better-focused educational intervention programs. 

## 2. Materials and Methods

This cross-sectional study received ethical approval from the university’s ethics committee (AU-LK-20180724). Participants were enrolled after signing a consent form. The study used a quantitative structured questionnaire to assess parental attitudes, knowledge, and behavior associated with sleep environments. Interviews were conducted at the parents’ homes during one meeting (July–August 2018). The interviewers were third-year undergraduate students trained to function as health promotion agents as part of a course named “Health promotion as a behavioral change at the individual level” taught at the Department of Health Systems Management, Faculty of Health Science, Ariel University.

### 2.1. Sample Description

A convenience sample of Israeli parents of infants (0–1 year of age) were recruited by the students. Each student had to locate 3 parents of infants from their hometown who agreed to take part in the study. They were contacted in a park near their residence where informed consent was obtained. The sample encompassed 335 parents (mean age 29.4, SD = 5.1, range 20–48 years old; 42 fathers and 293 mothers). The wide-ranging social background of the students resulted in a sample of parents from 59 different locations in Israel, which approximates the diversity of the Israeli population. Demographic data, family history, and clinical data were collected. The inclusion criteria were parents of infants (until 1 year old) who had a healthy baby, who had the cognitive ability to answer a questionnaire, whose custody of their baby belonged to them, who lived in their own home, and who signed a consent to participate in the study. Exclusion criteria were parents of infants with a respiratory problem, parents who had cognitive problems, parents of infants who did not have custody of the infant or did not live in their own home as a nuclear family, and parents who did not agree to participate in the study.

Demographic data of the parents are presented in Table 1. Regarding the infant, the mean pregnancy duration was 39 weeks (SD = 2.0), with almost all described as regular pregnancy (97.6%, only 8 were after IVF) and 96.1% single birth. Half of the newborns were female (50.2%), 72.3% were breastfed, and the mean weight was 3.1 kg (SD = 5.7). A total of 99.4% of the participants described their baby as healthy. 

### 2.2. Research Tool

The study consisted of a self-reported structured questionnaire delivered in Hebrew to the babies’ parents. The 11-page questionnaire included basic questions about the delivery, the newborn’s health, parental knowledge of the updated safe sleep recommendations/guidelines for babies, attitudes and parental behaviors, and sociodemographic questions. The questionnaire created by the researchers was validated by a pilot study of 10 parents. 

### 2.3. Variables Collected

Sociodemographic variables included gender (male, female); year of birth (fill up lines); ethnicity—participants were asked, “What ethnic background best describes you?,” and this question had 8 response values: 1. Ashkenazi/Western Europe, 2. Sephardi/Mizrahi, 3. Eastern Europe/Soviet Union/Russia, 4. Ethiopian, 5. Mixed origins, 6. Arab Israeli, 7. Druze, 8. Bedouin, and 9. Other (values 1–4 were included in the Jew category, values 6–8 were included in the Arab category, and values 5 and 9 were recoded as missing); income—participants were asked their opinion regarding their family’s average monthly income (this question had 5 response values: 1. Well above average, 2. Above average, 3. Average, 4. Below average, and 5. Far below average; values 1 and 2 were combined (above average) as well as 4 and 5 (below average)); religiosity—participants were asked to define their religiosity level, with values as follows: 1. Secular, 2. Traditional, 3. Religious, 4. Ultra-Orthodox, and 5. Other; country of birth—participants were asked to mark whether they were born in Israel or another country; parental education—participants were asked about the highest level of education they had reached (for mother and father separately), with values of 1. Less than eighth grade, 2. Graduated from ninth grade but did not continue high school, 3. High school but without graduation or matriculation certificate, 4. High school graduation but no other studies, 5. Special work training after high school, 6. Bachelor’s degree but without graduating, 7. Graduation with a bachelor’s degree, 8. Master’s degree and more, 9. I do not know, and 10. Other (values 1–4 combined the categories of high school or less, values 5–8 combined higher education, and values 9 and 10 were recoded as missing); mother smoking (yes, no, not relevant); mother smoking during pregnancy—mothers were asked whether they had smoked during pregnancy (values were 1. No, 2. 1–3 cigarettes a day, and 3. More than 4 cigarettes a day). Questions related to parental knowledge are described in Table 2, with the appropriate response highlighted. Parental attitudes are presented in the next section and in Table 3. Questions related to actual parental practices are described in Table 4, including a description of responses deemed in compliance with the current recommendations and those not. 

#### Parental Attitudes

We used a questionnaire to assess the “internal locus of control”, which measured the extent to which participants agreed that their baby’s safety depended on their own actions, control, or responsibility. The questions were based on the GLOC—God Locus Of Control questionnaire [14] and were appropriately adjusted (written in the Hebrew female language, and replace the wording of “health” with “baby safety.” Participants were asked to mark the extent to which they agreed with statements on a 1–5 Likert scale (1—very much agree, 5—very much disagree). Six statements (e.g., I am a person committed to the safety of my baby/My baby’s safety is in my complete control/I will do everything for my baby’s safety) were reversed and scaled (9–15; Cronbach’s alpha = 0.78), with higher scores expressing a higher internal locus of control.

We used a questionnaire to assess the “external locus of control” [14], which measured the extent to which participants agreed that their baby’s safety did not depend on their own actions, control, or responsibility, but rather on “external” factors. Participants were asked to mark the extent to which they agreed with statements on a 1–5 Likert scale (1—very much agree, 5—very much disagree). Six statements (for example, I hardly care how my baby falls asleep as long as he falls asleep/If something bad happened to my baby it’s just a matter of misfortune/Most of the bad things that happen to my baby are out of my control) were reversed and scaled (5–24; Cronbach’s alpha = 0.63), with higher scores expressing a higher external locus of control.

To assess parental attitude towards the need for more information about safe sleeping for their baby, participants were asked to mark the extent to which they agreed with various statement on a 1–5 Likert scale (1—very much agree, 5—very much disagree). There were three statements (e.g., I know the recommendations well and need no further information). The items were reversed and scaled (3–15; Cronbach’s alpha = 0.74), with higher scores expressing less need for more information.

Distrust recommendations measured the extent to which participants expressed distrust of the recommendations for a baby’s safe sleep. Participants were asked to mark the extent to which they agreed with statements on a 1–5 Likert scale (1—very much agree, 5—very much disagree). Seven statements (for example, I know what the recommendations are, but I don’t want to follow them/My baby just does not fall asleep according to the recommendations, so they are irrelevant to me/The recommendations for safe sleep are unfounded, and are constantly changed) were reversed and scaled (6–30; Cronbach’s alpha = 0.82), with higher scores expressing greater distrust.

Commitment to recommendations measured the extent to which participants were committed to complying with recommendations for a baby’s safe sleep. Participants were asked to mark the extent to which they agreed with statements on a 1–5 Likert scale (1—very much agree, 5—very much disagree). Seven statements (for example, It is very important for me to follow the recommendations for baby’s safe sleeping and I ask others to do the same while they take care of my baby/If someone were to teach me how to properly lay a baby, I would follow the recommendations/I will follow the recommendations even if I do not understand them) were reversed and scaled (6–30; Cronbach’s alpha = 0.79), with higher scores expressing greater commitment.

### 2.4. Statistical Analysis

Analysis was performed using the SPSS 25 statistical package. Initial analysis of descriptive statistics included frequencies of sociodemographic, knowledge, and behavioral variables. Chi square analysis for independent variables was used to assess unadjusted associations between gender, ethnicity, and parental experience and each dependent variable. Attitude scales were created after reliability tests with Cronbach’s alpha above 0.6. Means of parental attitudes were compared between independent groups (gender, ethnicity, and parental experience) using *T*-tests. A logistic regression was run to predict the outcome variable of baby’s sleep position.

## 3. Results

### 3.1. Knowledge

Table 2 presents parental knowledge of safe sleep recommendations for babies with respect to eight questions, by gender, ethnicity, and parental experience (whether they had less or more than two children) and the total knowledge score. Data are presented in percentages, chi-square significant outcomes are marked by asterisks, and correct answers are bolded. Most parents (82.2%) knew that they needed to lay their baby on their back, about two thirds (65.3%) of parents knew the correct answer with regard to the baby’s position while awake, and only 62.5% of participants knew that the baby should be put to sleep on their back during the daytime. Only 37.3% of the parents knew that breastfeeding is a protective factor against SIDS and only 41.4% knew that crib bumper pads around the baby’s head should not be used in the infant’s bed. Total knowledge score was higher for women (56.3%) than for men (26.6%; *p* < 0.001). Knowledge did not significantly vary by ethnicity or parental experience. A comparison of first-time parents versus parents with at least one other child did not reveal any significant differences (not presented in the table).

### 3.2. Attitudes

Table 3 presents parental attitudes regarding safe sleep positioning. The results of each series of questions are quoted as the mean with standard deviation (SD) in parentheses. The table shows no significant differences by gender with respect to all the attitude scales. There were significant differences between parents of different ethnic backgrounds: Arab parents had higher mean scores for questions indicating an internal locus of control and questions indicating an increased commitment to recommendations (mean = 14.7; SD = 0.6 and mean 29.3; SD2.7, respectively) in comparison to Jews (mean = 14.5; SD = 0.9 and mean 26.9; SD 4.7, respectively) (*p* < 0.05 and *p* < 0.001, respectively). Experienced parents with three children or more had higher mean scores (mean = 10.8; SD = 2.3) for “no need for information” compared to less experienced parents (mean = 10.1; SD = 2.3; *p* < 0.001). A comparison of first-time parents versus parents with at least one other child did not reveal any significant differences (not presented in the table).

### 3.3. Behaviors

Table 4 presents parental behaviors and practices. Parents most frequently complied with the following recommendations: avoidance of bed sharing with the baby (92.2%) and using only a closely fitted sheet on a firm mattress (89.9%). However, only 22.1% of parents reported fastening the blanket under the mattress and only 33.9% reported not using bumpers in the crib. 

An additional logistic regression analysis (Table 5) predicts the adjusted odds ratios (AORs) (with confidence intervals) for not placing their baby to sleep on their back for each parental response. Outcomes show that the risk of the baby not being placing to sleep on their back was higher if the parents were older than the mean of 29 years (AOR = 0.36; 95% CI 0.16–0.82; *p* < 0.05), if the father smoked (AOR = 2.66; 95% CI 1.12–6.33; *p* < 0.05), if the parents did not trust the recommendations (AOR = 4.03; 95% CI 1.84–8.84; *p* < 0.001), if they were not committed to the recommendations (AOR = 2.83; 95% CI 1.21–6.60; *p* < 0.001), and if they shared a bedroom with the baby (AOR = 0.38; 95% CI 0.17–0.88; *p* < 0.05). The significant variables in this model accounted for 40.7% of the explained variation. There was no association with socioeconomic status. 

## 4. Discussion

This study provides a unique perspective about current knowledge levels, attitudes, and practices regarding infants’ safe sleep among Israeli parents. It sampled 335 parents from 59 different communities in Israel, from north to south, consisting mostly of mothers and Jews (87% and 83%, respectively). The sample is only partially representative of Israel’s population, which, according to Central Bureau of Statistics 2019 data, is comprised of 74% Jews, 21% Arabs, and 5% others [15]. However, the fact that the parents resided in 59 different communities all over Israel indicates that the sample provides a good representation of Israeli parents of diverse communities and ethnic groups, as well as good insights into their safe sleep knowledge, perceptions, and implementation of the recommendations. The study’s findings will help health care providers identify behavioral barriers and issues that prevent the adoption of parental safe sleep behaviors and thus develop better-focused educational intervention programs. 

Looking at parental knowledge, despite extensive and ongoing “Back to Sleep” and “Safe to Sleep” publicity campaigns in Israel [12], only 82% of those sampled knew they had to put the baby to sleep on their back; however these figures are much better than those from Brazil [16], where 82% of women stated that the correct sleeping position for a baby is the lateral or prone position. Knowledge plays an important role in behavior changes [10,11,12], and just as studies demonstrate a lack of knowledge among parents, a systematic review of the literature showed that no studies reported complete adherence to the recommendations [17]. Adherence with safe sleep practices was poor despite knowledge of the American Academy of Pediatrics recommendations by adolescent mothers, whose babies are at increased risk. These mothers expressed beliefs and instincts that infants were safe in various unsafe sleep environments [18]. This was true also for other vulnerable parents. For instance, a study of low-income parents in Missouri found that although most participants were familiar with the recommendations, there was a lack of understanding about why they are necessary [19]. The decision-making process regarding infant safe sleep practices is complex; therefore, it is important for parents to understand what the evidence is in support of these recommendations.

It is also important for parents to understand the reasons why these recommendations are suggested (e.g., infants’ neck musculature may not be strong enough or the infant may not yet be coordinated enough to turn their head away from the mattress, increasing risk of suffocation). Understanding the reasons instead of just providing the recommendations would be a better way to increase compliance. Our sample indicates that there is a significant knowledge shortfall regarding safe sleep recommendations, especially those concerning supine sleep position during the day, use of bumper pads in a crib, loose beddings, and breastfeeding. This shortfall is less pronounced in mothers, who scored significantly higher than fathers, but no difference was found between Jews and Arabs or between experienced and less experienced parents. 

Looking at parental attitudes, Arab parents scored higher on questions indicating commitment to the baby’s safety and on questions indicating acknowledgement that the baby’s safety depends on the parents’ own actions. This might explain the previous finding of a higher compliance with “back to bed” sleeping position among Arabs compared to Jews [12]. Although knowledge scores were not related to level of experience, less experienced parents felt that they needed more information about safe sleep. Responses covering parents’ actual behavior demonstrated mixed results: Consistent with established safe sleep recommendations, 92% of the sampled parents reported avoidance of bedsharing and 89% reported using a firm mattress and fitted sheet. Our data indicate that although only 66.9% of the parents reported putting their baby to sleep in the supine position, this is significantly better than the 33% reported in a prior survey [5]. The data also indicate that the rate of bed sharing among Israeli parents was very low, whereas in the USA 61.4% of mothers reported bed sharing [20]. At the same time, our data indicate that room sharing has become more prevalent (73%) than in a prior survey [5]. However, the use of soft bedding (78%), placing bumper pads in the crib, and keeping the room temperature too warm (only 56% kept the room temperature between 22 and 23 °C) are still prevalent practices among Israeli parents, despite efforts to increase public awareness of the risks. The only variable significantly associated with the prediction of parents putting the baby to sleep prone was room sharing. This association could be the result of convenience and not related to safety considerations. If this is the case, it may also suggest that these parents think that if the baby is close to them, it may protect them and thus compensate for the risky behavior of prone sleeping (risk compensation phenomenon).

In Israel, targeted educational intervention has yielded considerable progress in reducing the incidence of infants placed to sleep in adult beds [5]. An option to rent a cot is available and utilized in this country and may have contributed to behavioral changes that improve safe sleep practices. Additionally, in the logistic regression analysis we found that knowledge does not necessarily predict behavior and that older parents are more likely to put the baby to sleep in a non-supine position. Unexpectedly, the study also found an association between room sharing and non-supine sleep position of the baby. A possible explanation as to why one guideline is followed and another ignored might be due to a compensation factor: The parent may believe room sharing compensates for the non-supine position. It is also possible that the parent adheres only to the recommendation that is easier to implement or is convenient and is unrelated to safety concerns.

Our study further indicated that older parents and fathers who smoke tend not to trust, and are not committed to, the recommendations; that a smoking father is 2.6 times more likely to put the baby to sleep in the non-supine position than a nonsmoking father; and that there is no such difference among mothers, as most of them do not smoke. These findings indicate the need to specifically develop interventions to address fathers. Although our study indicates improved adherence rates to the guideline of back sleeping in comparison to earlier surveys [5], prone placement to sleep (especially during the day) and an unsafe sleep environment (use of bed or crib bumper pads and loose bedding) are still prevalent among Israeli parents despite efforts to increase public awareness of the risks. Health care professionals need to continue to promote awareness amongst parents of the modifiable risk factors associated with SIDS and to ensure that all health care professionals model these practices.

Potential limitations of this study include validating the questionnaire on 10 subjects. This pilot included parents of infants who were part of the target population. The questions were understood correctly, and no significant changes in the questionnaire were required following the pilot phase. In addition, its reliance upon self-reporting by parents may not accurately reflect actual behavior due to a fear of being judged by the interviewer or social desirability. Since knowledge of some recommendations was quite satisfactory, it is possible that the reported behavior may not always reflect the actual practices adopted by the parents. 

## 5. Conclusions

This study indicates that specific infant care practices must be more clearly understood by the caregivers to close the gap between knowledge of and compliance with SIDS risk factor guidelines and safe sleep practices. Therefore, educational programs should target fathers and other caregivers and additional ways should be developed to increase trust in the guidelines that aim to reduce the risk of sudden death. Although in the initial years the Israeli campaign was focused on sleep position, now it is essential to promote a safe sleep environment for infants. This study can direct us to what we need to focus on going forward: developing new strategies to improve parents’ belief in the recommendations (e.g., training health educators and using home visits to augment knowledge; using face to face meetings to demonstrate safe sleep practices at the baby’s home). 

## Figures and Tables

**Table 1 children-08-00803-t001:** Sample description.

Variables	N	Percent	
Total	335	100	
Gender	Mothers	293	87.5
	Fathers	42	12.5
Ethnicity	Jews	279	83.8
	Arabs	49	14.7
Income	Above average	96	29.4
	Average	179	54.9
	Below average	51	15.6
Religion	Secular	95	28.6
	Traditional	79	23.8
	Religious/Orthodox	157	47.3
Country of birth	Israel	259	78.7
	Other country	70	21.3
Mother education	High school or less	42	12.7
	Higher education	282	85.2
Father education	High school or less	63	19.3
	Higher education	256	78.2
Mother smoking	No	305	92.1
	Yes	23	6.9
Mother smoking during pregnancy	No	312	97.5
	Yes, 1–3 cigarettes a day	3	0.9
	Yes, 4 cigarettes a day or more	5	1.6
Father smoking	No	242	72.7
	Yes	91	27.3
Number of kids	1	141	43.9
	2	83	25.9
	3 or more	90	28

**Table 2 children-08-00803-t002:** Distribution of parental knowledge regarding safety sleeping for babies (%).

Knowledge Questions	^a^Answers	All Sample	Gender		Ethnicity		Parental Experience	
			Females *n* = 293	Males *n* = 42	Jews *n* = 279	Arabs *n* = 49	≤2 Kids *n* = 231	≥3 Kids *n* = 90
1. How should a baby be put to bed at night?	1. On the stomach	7.5	7.2	9.5	7.6	16	7.9	6.7
	**2. On the back**	82.2	82.1	83.3	82.3	79.6	80.3	86.7
	3. On the side	6.9	7.2	4.8	6.1	12.2	8.7	3.3
	4. As convenient to the baby	3.3	3.4	2.4	4	0	3.1	3.3
2. How should a baby be put to bed during day sleep?	1. On the stomach	16.9	16.2	22	18.1	12.2	17	18
	**2. On the back**	62.5	63.8 *	53.7 *	61.4	67.3	62	61.8
	3. On the side	6.9	5.5	17.1	6.1	12.2	7	6.7
	4. As convenient to the baby	13.6	14.5	7.3	14.4	8.2	14	13.5
3. What is the recommended position when a baby is awake and under supervision?	**1. On the stomach**	65.3	69.2 **	38.1 **	67.7 *	51.0 *	64.1	68.9
	2. On the back	16.8	15.1	28.6	16.1	20.4	18.2	13.3
	3. On the side	2.4	1.7	7.1	1.4	8.2	2.2	2.2
	4. As convenient to the baby	15.6	14	26.2	14.7	20.4	15.6	15.6
4. Should crib bumper pads be used?	1. Yes, they are worthwhile to use for the baby’s head protection	50.8	49	63.4	48.2	65.3	49.1	51.1
	**2. Should not be used for suffocation prevention**	41.4	43.5	26.8	43.9 ^a^	26.5 ^a^	43	40
	3. I don’t know	7.8	7.5	9.8	7.9	8.2	7.8	8.9
5. What is the best way to cover your baby during sleep?	1. In a heavy, free blanket	3	2.8	4.9	1.4	12.2	3.9	1.1
	2. A thin, free blanket	43.5	41.4	58.5	46	28.6	41.9	43.8
	3. A thick, heavy blanket fastened under armpits and securely attached to the mattress	4.2	4.5	2.4	3.6	8.2	6.1	0
	**4. A thin blanket fastened under armpits and securely attached to the mattress**	49.2	51.4 *	34.1 *	48.9 ***	51.0 ***	48.0 *	55.1 *
6. How is breastfeeding and baby health and safety related?	1. No connection	4.5	5.2	0	4.7	4.1	4.8	4.4
	**2. Protective factor against SIDS and desirable if possible**	37.3	38.8 ^a^	26.8 ^a^	39.4	26.5	38.6	36.7
	3. Best nutrition for the baby and is not related to SIDS	58.2	56.1	73.2	56	69.4	56.6	58.9
7. Do you think that using a pacifier during bedtime…?	1. Increases the risk of death during sleep	8.7	7.9	14.3	7.6	16.3	9.1	7.9
	**2. Reduces the risk of death during sleep**	53.9	56.2a	38.1a	53.8a	57.1a	54.8	56.2
	3. I don’t know	37.3	35.9	47.6	38.6	26.5	36.1	36
8. Do you think that infant deaths never happen to healthy babies?	1. Agree	10.2	8.2	23.8	9.7	14.3	10	12.4
	2. I am not sure	29.7	30.2	26.2	30.5	24.5	29.9	28.1
	**3. Disagree**	60.1	61.5 **	50.0 **	59.9	61.2	60.2	59.6
**Total knowledge score**	**Higher knowledge (>5)**	52.8	56.3 ***	28.6 ***	55.2	42.9	55.8	56.7

Chi-square significant * *p* < 0.05, ** *p* < 0.01, *** *p* < 0.001, ^a^
*p* = 0.06–0.07; for differences groups of gender, nationality, and parental experience. **^a^**
**correct answer is bolded.**

**Table 3 children-08-00803-t003:** Mean distribution of parental attitudes regard baby’s safe sleeping.

Attitudes	Scales Values	All Sample M (SD)	Gender		Ethnicity		Parental Experience	
			Females M (SD)	Males M (SD)	Jews M (SD)	Arabs M (SD)	≤2 Kids M (SD)	≥3 Kids M (SD)
		*n* = 323–332	*n* = 281–290	*n* = 42	*n* = 269–277	*n* = 49	*n* = 231	*n* = 90
Internal locus of control	9–15; 15 = high internal	14.6 (0.9)	14.6 (0.9)	14.7 (0.6)	14.5 * (0.9)	14.7 * (0.6)	14.6 (0.9)	14.5 (0.8)
External locus of control	5–24; 24 = high external	11.2 (3.2)	11.2 (3.1)	11.4 (3.3)	11.1 (3.2)	11.8 (3.5)	11.2 (3.3)	11.6 (2.9)
Do not need any more information	3–15; 15 = less need	10.2 (2.3)	10.2 (2.3)	10.5 (2.7)	10.2 (2.4)	10.6 (2.2)	10.1 ** (2.3)	10.8 ** (2.3)
Distrust recommendations	6–30; 30 = highly distrust	13.8 (4.9)	13.8 (5.1)	13.9 (3.7)	13.9 (5.0)	13.2 (3.9)	13.7 (5.0)	13.8 (4.6)
Commitment to recommendations	11–35; 35 = high commitment	27.3 (4.6)	27.1 (4.6)	28.4 (4.6)	26.9 ** (4.7)	29.3 ** (2.7)	27.4 (4.5)	27.1 (4.7)

Significant * *p* < 0.05, ** *p* < 0.001; for differences groups of gender, nationality, and parental experience.

**Table 4 children-08-00803-t004:** Distribution of parental practice considering recommendation for baby’s safe sleeping.

	Practice Type	Behavior According to Recommendations			Misbehavior Recommendations			n
		Description	Value/s	%	Description	Value/s	%	
1	What bedding did your baby sleep on last night?	A rigid mattress for infants with a tight sheet for the mattress and a baby-sized mattress	1	89.9	A rigid mattress without a tight sheet/A soft mattress with a tight sheet/A sheet too large for the size of the bed that is not tight on the mattress/A thin blanket under the baby/Thick blanket under the baby.	2–6	10.1	317
2	If the baby was covered with a blanket, the blanket was:	Tucked and fastened under the mattress	2	22.1	Free to move with baby movements	1	77.9	285
3	Did the baby sleep on a pillow last night?	No	1	86.1	Yes, just head on the pillow/Yes, the whole body on the pillow	2, 3	13.9	330
4	Does the crib or the baby bed in which they sleep usually have a head shield for the bed?	No	2	33.9	Yes	1	66.1	333
5	Are there toys in the bed while sleeping?	No	1	82.3	Yes, soft stuffed animals/Yes, different toys	2, 3	17.7	333
6	What is the position where you usually lay your baby to bed?	Lie on their back	3	66.9	Lies on stomach facing down/Lies on stomach with face to the side/Lying on the side	1, 2, 4	33.1	335
7	In what room does the baby normally sleep?	In the parents’ room	2	73.3	In a room of their own/In a room with other children	1, 3	26.7	330
8	Bed sharing?	No	1	92.2	Yes, adults/Yes, kids	2, 3	7.8	320
9	Do you offer a pacifier to the baby during bedtime?	Yes Always/Yes Sometimes	1, 2	81.7	No	3	18.3	334
10	Does your baby sleep in their stroller for nighttime sleep?	No	1	86.2	Yes, rarely/Yes, sometimes/Yes, most of the time	2–4	13.8	333

**Table 5 children-08-00803-t005:** Outcomes of logistic regression—adjusted odds ratios (AORs) for associations between baby sleep position and other study variables.

Measures		Ref—Back Only	Sleep Position—Not on the Back		
Sociodemographic			AOR	95% CI	
Gender	(0 = females, 1 = males)	1.00	0.61	0.18	1.99
Age	(0 = older, 1 = younger)	1.00	**0.36 ***	0.16	0.82
Mother education	(0 = high, 1 = low)	1.00	0.77	0.35	1.72
Father education	(0 = high, 1 = low)	1.00	1.08	0.49	2.40
Ethnicity	(0 = Jews, 1 = Arabs)	1.00	0.44	0.14	1.39
Income	(0 = high, 1 = low)	1.00	1.14	0.40	3.21
Religion	(0 = secular, 1 = religious/traditional)	1.00	0.83	0.32	2.12
Mother smokes	(0 = no, 1 = yes)	1.00	1.35	0.26	6.95
Father smokes	(0 = no, 1 = yes)	1.00	**2.66 ***	1.12	6.33
Baby’s age	(0 = >6 months, 1 = <6 months)	1.00	1.50	0.72	3.10
Knowledge score	(0 = high, 1 = low)	1.00	1.70	0.77	3.76
Internal LOC	(0 = high, 1 = low)	1.00	0.48	0.19	1.22
External LOC	(0 = low, 1 = high)	1.00	1.21	0.54	2.70
Information	(0= do not need, 1 = need)	1.00	1.06	0.51	2.21
Trust recommendations	(0 = trust, 1 = do not trust)	1.00	**4.03 ****	1.84	8.84
Commitment to recommendations	(0 = commit, 1 = not committed)	1.00	**2.83 ***	1.21	6.60
Blanket	(0 = fastened, 1 = free to move)	1.00	0.46	0.17	1.23
Bedding	(0 = mattress stiff sheets fastened, 1 = else)	1.00	2.62	0.77	8.90
Room sharing	(0 = in parents’ room, 1 = else)	1.00	**0.38 ***	0.17	0.88
Sleep position	(0 = back only, 1 = not on the back)				
Sig		*p* = 0.000			
Nagelkerke R^2^		40.7%			
*n*		195			

Significant for exp B * *p* < 0.05, ** *p* < 0.001. Significant results are bolded.

## Data Availability

The data presented in this study are available on request from the corresponding author.

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
