# Peer review of "Infants’ Sleep: Israeli Parents’ Knowledge, Attitudes and Practices"

_children, 2021, doi:10.3390/children8090803_

Round 1
Reviewer 1 Report
The authors examined the knowledge, attitudes, and practices of Israeli parents associated with infant sleep environments. Given that the authors note less than half of Israeli parents place their infants on their back to sleep, studies that examine why parents are not engaging in safe sleeping practices in Israel are critical to reducing SIDS rates. However, there are a number of methodological concerns that need to be addressed before the potential contributions of the study can be evaluated properly. Below is a summary of the strengths and weaknesses of the study as well as a description of specific concerns/edits noted throughout the manuscript.
Strengths include a large, diverse sample of parents from across Israel that includes assessments of infant sleep safety attitudes, beliefs, and practices. Data such as these are needed to improve SIDS rates and healthy sleep for infants.
Weaknesses While the topic is one worthy of study, the methodology needs to be more clearly described, and the current information provided does not allow for confidence in the findings. The questionnaire was developed for this study and authors indicate that it was validated with a pilot study of 10 participants (were these experts or persons from the target population?), but there is no information provided that shows the process or validity indicators (e.g., face, criterion, construct).
Specific recommended edits/considerations:
- In the abstract it states “to find intervention programs” when really the research is done to inform the development of effective intervention programs.
- Be consistent with the use of first person. For example, in the abstract it reads “Researchers visited homes…We used SPSS…”
- What is the SIDS rate of Israel? That information would strengthen the case for the study.
- Line 38 the word western should be capitalized.
- Line 47 should “smoing” be smoking?
- Suggest that line 49 “Breastfeeding is recommended for at least two months” be added to the list of evidence-based recommendations because as is, it seems out of place.
- Verb tenses shift often throughout the introduction.
- Line 62 indicates that gender plays a role in determining sleep position, but the direction of that finding is not clear.
- Line 65 I suggest that you take out “the” and add “safe sleep” prior to the word recommendations.
- Line 67 you use “they” to refer to the parents but two words later the word they is referring to strategies. This needs to be clarified for the reader. In that same line, “their actual practice” might read better if it was “and what their actual practices are.”
- Lines 68-69 suggest removing “on” prior to identifying and adding “of these” prior to the word variables.
- Line 70 there is an extra space before the comma.
- Line 72 remove the word thus.
- Line 77 indicates that the structured questionnaire analyzed survey results. This sentence needs to be reworded. I suggest something such as “The study used a quantitative structured questionnaire to assess parental attitudes, knowledge and behavior associated with sleep environments.” Or something similar.
- Line 87 includes information that was presented in line 84. Redundancy should be removed.
- The wording needs to changed in lines 87-89. For example, “the sample does not pretend” is not grammatically correct. Making a more robust sentence out of the two would allow for better reading flow. I also suggest tweaking the wording in lines 89-92 as well.
- Line 90, the C in criteria should be lower case.
- Mean weight was 3.1 pounds? This information does not seem correct. Please provide the metric and/or double check that number.
- There is missing information in Table 1. For example, did all 335 participants provide age? I would suggest that the information provided in the text not be duplicated in the tables. The table could be simplified if the min-max and mean columns were removed since only one variable is recorded there.
- More specific information is needed regarding the structured questionnaire. Example questions, at a minimum, should be provided and the reader should know how many questions there were per construct before that information is presented in the results section. For example, how many questions targeted the attitudes of parents? Given the information in Table 1, it is expected that sociodemographic questions included income within certain ranges. Those specifics are critical for the readers to know to accurately evaluate the study diversity in sample and statements that SES was not related to the variables of interest.
- While the tables provide some information about sociodemographic variables, how the categories were established is not clear. For example, how was average established? Were participants asked to select which category they feel into or did the researchers provide income ranges?
- In the Parental Attitudes section (line 118) it says that the questionnaire was adjusted appropriately. The reader should not just take the researcher’s word for what is appropriate. There needs to be further information provided about the changes made to the questionnaire.
- The alpha for the External locus of control measure is low and would suggest reduced reliability in these findings.
- Check the grammar on lines 152-155.
- Line 168 should say “or” instead of “and”.
- Table 2: Not all of the questions are written in question form (need consistency) and there are grammatical errors (e.g., #5 doesn’t start with a capital letter and is not spaced from the number).
- Why the division in parental experience at 2? Why not use one child vs. multiple children? There should be justification from the literature or from a statistical perspective for this grouping.
- Line 281 has extra spaces.
- Was all of the data collected in one semester or over multiple iterations of the course? Were the interviews conducted verbally and always in the home or did participants fill out the survey?
- Some of the information reported in the discussion would fit better in the results section.
- Would including reasons why these recommendations are suggested (e.g., infants’ neck strength may not be strong enough or the infant may not yet be coordinated enough to turn their head away from the mattress, increasing risk of suffocation) instead of just providing the recommendations be one way to increase compliance? Adult education literature would suggest that this is an important component that, from this review, is unclear if it is part of the current messaging.
Author Response
Reviewer 1:
Comments and Suggestions for Authors
The authors examined the knowledge, attitudes, and practices of Israeli parents associated with infant sleep environments. Given that the authors note less than half of Israeli parents place their infants on their back to sleep, studies that examine why parents are not engaging in safe sleeping practices in Israel are critical to reducing SIDS rates. However, there are a number of methodological concerns that need to be addressed before the potential contributions of the study can be evaluated properly. Below is a summary of the strengths and weaknesses of the study as well as a description of specific concerns/edits noted throughout the manuscript.
Strengths include a large, diverse sample of parents from across Israel that includes assessments of infant sleep safety attitudes, beliefs, and practices. Data such as these are needed to improve SIDS rates and healthy sleep for infants.
Weaknesses While the topic is one worthy of study, the methodology needs to be more clearly described, and the current information provided does not allow for confidence in the findings. The questionnaire was developed for this study and authors indicate that it was validated with a pilot study of 10 participants (were these experts or persons from the target population?), but there is no information provided that shows the process or validity indicators (e.g., face, criterion, construct).
Authors: Thank you so much for your clear comments and time dedication. The questionnaire was validated by parents of infants who are part of the target population, they understood the questions easily, and no significant changes were required following the pilot phase. This was added as a limitation of the study. However we would like to mention that a previous version of the questionnaire was used for internal purposes by a healthcare service in order to monitor mothers knowledge and practices. We have addressed the Reviewer’s concerns and carefully reviewed the entire paper. We hope that the Reviewer will find the revised manuscript suitable for publication.
Specific recommended edits/considerations:
- In the abstract it states “to find intervention programs” when really the research is done to inform the development of effective intervention programs.
Authors: The confusing end of the sentence was omitted from the abstract.
- Be consistent with the use of first person. For example, in the abstract it reads “Researchers visited homes…We used SPSS…”
Authors: Changed as requested.
- What is the SIDS rate of Israel? That information would strengthen the case for the study.
Authors: This information was added to the text as follow: “In Israel, the Central Bureau of Statistics publishes data for SUID (Sudden Unexpected Infant Death) instead of SIDS due to a lack of routine post mortem or death scene investigation. Rates in the last decade ranged from 0.6-0.23 per 1000 live births. The most recent available reports for the years 2016 and 2017 indicated 40 and 53 cases of SUID respectively.”
- Line 38 the word western should be capitalized.
Authors: Done as suggested.
- Line 47 should “smoing” be smoking?
Authors: Corrected.
- Suggest that line 49 “Breastfeeding is recommended for at least two months” be added to the list of evidence-based recommendations because as is, it seems out of place.
Authors: Done as suggested.
- Verb tenses shift often throughout the introduction.
Authors: We have carefully reviewed the introduction section and have made few changes of verb tenses where needed.
- Line 62 indicates that gender plays a role in determining sleep position, but the direction of that finding is not clear.
Authors: We added information regarding the direction of this association: “Gender was also found to play an important role in determining sleep position. Male caregivers demonstrated less knowledge of the recommendations regarding infant sleep safety and they adhered less to all of the safe sleep recommendations”.
- Line 65 I suggest that you take out “the” and add “safe sleep” prior to the word recommendations.
Authors: Done as suggested. Thank you.
- Line 67 you use “they” to refer to the parents but two words later the word they is referring to strategies. This needs to be clarified for the reader. In that same line, “their actual practice” might read better if it was “and what their actual practices are.”
Authors: We had clarified the text as suggested. Thank you.
- Lines 68-69 suggest removing “on” prior to identifying and adding “of these” prior to the word variables.
Authors: Done as suggested. Thank you.
- Line 70 there is an extra space before the comma.
Authors: Removed.
- Line 72 remove the word thus.
Authors: Removed.
- Line 77 indicates that the structured questionnaire analyzed survey results. This sentence needs to be reworded. I suggest something such as “The study used a quantitative structured questionnaire to assess parental attitudes, knowledge and behavior associated with sleep environments.” Or something similar.
Authors: Done as suggested. Thank you.
- Line 87 includes information that was presented in line 84. Redundancy should be removed.
Authors: Removed.
- The wording needs to changed in lines 87-89. For example, “the sample does not pretend” is not grammatically correct. Making a more robust sentence out of the two would allow for better reading flow. I also suggest tweaking the wording in lines 89-92 as well.
Authors: Reviewed and edited where needed.
- Line 90, the C in criteria should be lower case.
Authors: Done as suggested.
- Mean weight was 3.1 pounds? This information does not seem correct. Please provide the metric and/or double check that number.
Authors: kg was added.
- There is missing information in Table 1. For example, did all 335 participants provide age? I would suggest that the information provided in the text not be duplicated in the tables. The table could be simplified if the min-max and mean columns were removed since only one variable is recorded there.
Authors: Mean and min-max were removed from the table and the information was left only in the text.
- More specific information is needed regarding the structured questionnaire. Example questions, at a minimum, should be provided and the reader should know how many questions there were per construct before that information is presented in the results section. For example, how many questions targeted the attitudes of parents? Given the information in Table 1, it is expected that sociodemographic questions included income within certain ranges. Those specifics are critical for the readers to know to accurately evaluate the study diversity in sample and statements that SES was not related to the variables of interest.
Authors: Example questions were provided in the next section of variables description (section 2.3). Regarding sociodemographic questions, we added detailed description in the text as requested.
- While the tables provide some information about sociodemographic variables, how the categories were established is not clear. For example, how was average established? Were participants asked to select which category they feel into or did the researchers provide income ranges?
Auhors: We added detailed information in the text as requested.
- In the Parental Attitudes section (line 118) it says that the questionnaire was adjusted appropriately. The reader should not just take the researcher’s word for what is appropriate. There needs to be further information provided about the changes made to the questionnaire.
Authors: We added detailed information in the text as requested: Written in Hebrew female language and replace wording of “health” with “baby safety”.
- The alpha for the External locus of control measure is low and would suggest reduced reliability in these findings.
Authors: The Reviewer may be right, since Alpha Cronbach 0.63 is not particularly strong, but it is strong enough to allow the creation of a complex measure. The ability to withdraw items in order to increase the alpha value was tested using the "alpha if items are deleted" function and in the manuscript the highest score that could be obtained was presented.
- Check the grammar on lines 152-155.
Authors: Checked and edited accordingly.
- Line 168 should say “or” instead of “and”.
Authors: Corrected as suggested.
- Table 2: Not all of the questions are written in question form (need consistency) and there are grammatical errors (e.g., #5 doesn’t start with a capital letter and is not spaced from the number).
Authors: Questions were changed to show consistency as they appeared in the questionnaire. Grammar error was corrected.
- Why the division in parental experience at 2? Why not use one child vs. multiple children? There should be justification from the literature or from a statistical perspective for this grouping.
Authors: Thank you. As the Reviewer pointed out, the rational for this division derived both from theoretical perspective and statistical perspective. We checked the two options of comparisons: 1. First time parents versus parents with one older child or more. 2. Less experienced parents who have only one older child versus more experienced parents who have at least two older siblings. Theoretically, the more experienced parents are, the more differentiation is expected between groups, which is why we thought to check the second comparison as well.
Statistical analysis of both comparisons showed no significant differences between independent groups in the case of comparison 1, both in knowledge and in attitudes measures. Comparison 2 as shown in the manuscript (tables 2 + 3) resulted with two significant differences between groups. That was the reason for us to stay with the second comparison. As for the reviewer concern, we added the information regard the first comparison outcomes in the text at the result section.
- Line 281 has extra spaces.
Authors: Removed.
- Was all of the data collected in one semester or over multiple iterations of the course? Were the interviews conducted verbally and always in the home or did participants fill out the survey?
Authors: Information was added to the text: Interviews were conducted at the parents’ homes in one meeting during July-August 2018.
- Some of the information reported in the discussion would fit better in the results section.
Authors: The discussion suction does contain reports of findings but usually as a shorten repetition of the results, and to assist in the comparison to the literature.
- Would including reasons why these recommendations are suggested (e.g., infants’ neck strength may not be strong enough or the infant may not yet be coordinated enough to turn their head away from the mattress, increasing risk of suffocation) instead of just providing the recommendations be one way to increase compliance? Adult education literature would suggest that this is an important component that, from this review, is unclear if it is part of the current messaging.
Authors: Thank you. We agree and have added this point to the discussion.

Reviewer 2 Report
This is interesting and important cross-sectional questionnaire based study concerning assessmant of Israeli parents’ knowledge and attitudes towards practices promoting infants' safe sleep. Generally it is well designed and written paper. However it has a few minor flwas:
1. The aim of the study is not clearly define at the end of the Introduction. It has to be corrected and clearly stated.
2. Authors have to clearly define exclusion criteria used for parents' recruitation in chapter 2.1. Procedure of Parents' recruitment has to be describe much more clearlier.
3. Authors used self-designed questionnaire which was validated in a small group of 10 parents. This is the biggest flaw of this study. Therefore it has to be describe as a study limitation at the end of the Discussion.
4. The number of References is too small. I recommend to extend the number of cited papers in the Discussion, especially confront your findings with current findings in this topic of authors from all over the world. I think that min. number of references should be above 20.
Author Response
Reviewer 2:
Comments and Suggestions for Authors
This is interesting and important cross-sectional questionnaire based study concerning assessmant of Israeli parents’ knowledge and attitudes towards practices promoting infants' safe sleep. Generally it is well designed and written paper. However it has a few minor flwas:
1. The aim of the study is not clearly define at the end of the Introduction. It has to be corrected and clearly stated.
Authors: Thank you. Done as suggested.
Authors have to clearly define exclusion criteria used for parents' recruitation in chapter
Authors: exclusion criteria were added as requested.
2.1. Procedure of Parents' recruitment has to be describe much more clearlier.
Authors: We added more details regarding the recruitment method and wrote this more clearly in the text as requested.
Authors used self-designed questionnaire which was validated in a small group of 10 parents. This is the biggest flaw of this study. Therefore it has to be describe as a study limitation at the end of the Discussion.
Authors: Agreed. We added this as a limitation.
The number of References is too small. I recommend to extend the number of cited papers in the Discussion, especially confront your findings with current findings in this topic of authors from all over the world. I think that min. number of references should be above 20.
Authors: Thank you. We have added few more citations to the discussion.
